# Review of DSP Toxicity in Ireland: Long-Term Trend Impacts, Biodiversity and Toxin Profiles from a Monitoring Perspective

**DOI:** 10.3390/toxins11020061

**Published:** 2019-01-22

**Authors:** Rafael Salas, Dave Clarke

**Affiliations:** Marine Environment and Food Safety, Marine Institute, Rinville, Oranmore, H91 R673 Galway, Ireland; dave.clarke@marine.ie

**Keywords:** dinophysis, DSP, toxins, OA, DTX-2, PTXs

## Abstract

The purpose of this work is to review all the historical monitoring data gathered by the Marine Institute, the national reference laboratory for marine biotoxins in Ireland, including all the biological and chemical data from 2005 to 2017, in relation to diarrheic shellfish poisoning (DSP) toxicity in shellfish production. The data reviewed comprises over 25,595 water samples, which were preserved in Lugol’s iodine and analysed for the abundance and composition of marine microalgae by light microscopy, and 18,166 records of shellfish flesh samples, which were analysed using LC-MS/MS for the presence and concentration of the compounds okadaic acid (OA), dinophysistoxins-1 (DTX-1), dinophysistoxins-2 (DTX-2) and their hydrolysed esters, as well as pectenotoxins (PTXs). The results of this review suggest that DSP toxicity events around the coast of Ireland occur annually. According to the data reviewed, there has not been an increase in the periodicity or intensity of such events during the study period. Although the diversity of the *Dinophysis* species on the coast of Ireland is large, with 10 species recorded, the two main species associated with DSP events in Ireland are *D. acuta* and *D. acuminata*. Moreover, the main toxic compounds associated with these species are OA and DTX-2, but concentrations of the hydrolysed esters are generally found in higher amounts than the parent compounds in the shellfish samples. When *D. acuta* is dominant in the water samples, the DSP toxicity increases in intensity, and DTX-2 becomes the prevalent toxin. Pectenotoxins have only been analysed and reported since 2012, and these compounds had not been associated with toxic events in Ireland; however, in 2014, concentrations of these compounds were quantitated for the first time, and the data suggest that this toxic event was associated with an unusually high number of observations of *D. tripos* that year. The areas of the country most affected by DSP outbreaks are those engaging in long-line mussel (*Mytilus edulis*) aquaculture.

## 1. Introduction

The study of marine biotoxins is an important area of scientific research that is mainly driven by poisoning incidents in humans who consume bivalves. Diarrheic shellfish poisoning (DSP) was first recorded in the Netherlands in the 1960s, when cases of a form of gastroenteritis linked to mussel consumption were identified [1,2]. In Japan, a similar event occurred in the late 1970s [3,4]. 

The recording of DSP toxicity events in Ireland and the rapid development of the shellfish industry seem to have taken place in parallel. One of the first DSP toxicity events ever recorded in Ireland took place in 1984, which coincided with the start of marine biotoxin monitoring. DSP toxins were first detected by using a DSP rat bioassay (1990–1996) [2], by which samples of hepatopancreas tissue from shellfish were orally fed to rats and the consistency of the resultant faeces was rated from dry to liquid. In the 1980s, only summer closures were enforced; however, in 1994, toxins were detected by LC-MS/MS outside of the months during which there was commonly believed to be a greater risk of toxic events. DSP toxicity (OA equivalents) in mussel hepatopancreas reached a high of 13.5 µg∙g^−1^ in August of 1994, and dinophysistoxins-2 (DTX-2) concentrations remained above the regulatory limits until February of 1995 (>2 µg∙g^−1^). This toxic episode was associated with high cell densities of *Dinophysis acuta* (20,000 cells∙L^−1^) observed in Bantry Bay [5] and identified the need for a year-round biotoxin monitoring programme. The incident also highlighted that the rat bioassay was not a method fit for this purpose, because it was considered to be too subjective; as a result, the rat bioassay was eventually replaced by the more sensitive mouse bioassay.

The DSP mouse bioassay was used for several years (1997–2011) [3], but in 2000, chemical characterisation via LC-MS became more widely used [3,6]. In 2002, a hydrolysis step was introduced into the LC-MS/MS method to account for the total toxin equivalent of the main toxins and to allow for the detection of the acyl derivatives of okadaic acid (OA) and other toxins in that group [7]. During the 2000s, both the mouse bioassay and the LC-MS DSP method were run in parallel until the mouse bioassay was finally discontinued in 2011. Today, the European Union (EU) harmonised LC-MS/MS protocol [8] is used as the reference method. The main toxins responsible for DSP incidents in Ireland are OA [9], DTX-2, their hydrolysed esters [10] and the non-diarrheic pectenotoxin (PTX) group [6,11]. These polyether compounds are potent phosphatase inhibitors [12] that cause intestinal illness, except for PTXs, which are hepatotoxic [13]. In this work, only the chemical data produced through LC-MS/MS analysis from 2009 to 2017 for the OA and DTX-2 parent compounds and from 2011 to 2017 for their hydrolysed esters were used. 

The phytoplankton monitoring programme commenced in 1986, and originally, the main interest was in counting the high biomass phytoplankton species only. It was believed that many cells were needed in a sample to produce shellfish toxicity. Only a limited number of species were counted per sample, and limited data were recorded about the sample and how it was collected. This continued until 1995, when the first phytoplankton database, ‘Fytobase’, was developed by the Marine Institute. Fytobase was the first Microsoft Access database, and it was a definite improvement from the previous system. It included a standardised phytoplankton species list and unlimited space for species entry. Additionally, sample data collection, including the methodology employed, sample depth, and latitude/longitude coordinates, were recorded.

In 2002, Fytobase was superseded by the Harmful Algal Blooms database (HABs). HABs was decommissioned in 2018 and replaced by a new Windows-based system, HABs2. This new system differs from the previous one in that it is a fully automated, digitalised platform, which allows the Marine Institute to publish phytoplankton results in close to real time, and it is fully open to the public. The phytoplankton results can be plotted over a timeline to show the trends of the main toxic species, and these can be compared against the toxin results for the same area. These figures can give near real-time information and serve as an early warning system of an impending toxic event for the shellfish industry. 

The biological data used in this review were collected from all the shellfish and finfish production areas in Ireland between 2005 and 2017 as part of the monitoring programme. The datasets from this time period are much more comprehensive and reliable than previous ones, as the monitoring programme has been improving, culminating in the accreditation of the Utermöhl test method [14] in 2004 to the ISO standard of 17,025. The monitoring programme comprises samples from over 80–90 shellfish production sites weekly, although not all these sites are active year-round. 

The data for the purpose of this study was organised into larger geographical regions rather than individual sites or bays along the coast in order to obtain realistic values and achieve some proportionality among the regions. The rationale for this was to be able to objectively compare the coastal areas that are not as heavily influenced by shellfish production. For example, the eastern region along the Irish Sea only accounts for eight production areas as compared with the 64 production areas in the western region.

The observations of *Dinophysis* spp. in the water column in Ireland typically occur in the summer months and in the thermally stratified waters along the shelf front [15] in all the coastal locations, particularly in the southern and southwestern areas. Here, the oceanography can be quite complex, as the continental shelf is less than 100 m deep and can be highly stratified in the summer. *Dinophysis* populations can develop quite quickly and be advected into bays by prevailing winds. This advection depends on the position of the shelf front [5]. This exchange can be accentuated due to the orientation of the bays in the southwest relative to the prevailing winds [16]. 

The average sea surface temperatures for the western and southern waters of Ireland range between 8–10 °C in the winter and 14–17 °C in the summer [17] and tend to be several degrees higher compared with the eastern waters due to the Gulf stream current modulating the temperatures in western Ireland. In the winter, the Irish coastal waters are well mixed, with no differences between the bottom and surface temperatures, but in the summer, as the waters stratify, there can be a 5–6 °C difference between the surface and the bottom [18]. It is in the area between the coastal mixed waters and stratified offshore waters marked by a tidal front that *Dinophysis* populations are found. The transport mechanism in the southern and southwestern regions is well studied and explained by the Irish Coastal Current (ICC) [19]. This is a strong jet-like fast current moving in a clockwise direction around the south and southwest of Ireland, and it can be modulated by the shelf front due to wind forcing [16]. This transport mechanism is believed to be essential to the delivery of *Dinophysis* into these bays [20]. 

## 2. Results

### 2.1. Phytoplankton Records from the HABs Database

Between 2005 and 2017, a total of 25,595 phytoplankton samples were analysed, containing 346,186 phytoplankton records, of which 5315 were *Dinophysis*. During this period, 10 different species of the genus *Dinophysis* were observed in the samples (Table 1), which is a consistent measure of the diversity of this genus in Irish waters. The most observed species, without a doubt, were *Dinophysis acuminata* and *Dinophysis acuta*. The third most observed species after that was *Dinophysis tripos*, followed by a small number of observations for *D. caudata*, *D. norvegica*, *D. hastata*, *D. fortii*, *D. ovum* and *D. odiosa* and only one record of *D. nasuta*. 

Our data shows that the DSP outbreaks around the Irish coast fluctuated between years of high toxicity (1994 and 1995; 2000 and 2001; 2012, 2014 and 2015) and years of low or no toxicity (1997 and 1998; 2002, 2016 and 2017). The periods of high toxicity were associated with a high number of observations of *D. acuta* and *D. acuminata* and vice versa. Other *Dinophysis* species were also present in the water with *D. tripos*, which was the next most recorded species. Up to 10 different species of this genus were recorded by the monitoring programme during this period, albeit most of them only a handful of times, so their association to toxicity in Ireland is limited. 

Table 2 shows the normalised, estimated number of observations of *D. acuminata* and *D. acuta* from 1500 records in each geographical area. The southern region had the least total number of samples of all the areas during the studied period, with only 1500 samples collected between 2005 and 2017. The reason for this is that the southern region also had the smallest number of active sampling sites (5). Proportionally, 1937 samples were collected during the same period in the southwest, and 1621 samples were collected in the western region, which corresponded to an average of 40 and 32 samples per year, respectively (data not shown).

The data shows that the areas with the largest number of observations for both species were in the southern and southwestern regions in comparison to the Irish Sea (east and southeast) and the western and northwestern Atlantic areas. Moreover, the ratios between *D. acuminata* and *D. acuta* in the south and southwest were different than in other regions. In the northwestern and western regions, *D. acuminata* was clearly the most observed of the two species, with ratios of 4:1 and 5:1. In the east and southeast, this tendency was not as obvious, with *D. acuminata* still being the most dominant at ratios of 3:2 and 2:1, respectively. There was a much closer ratio of 4:3 in the southwestern region and a ratio favourable to *D. acuta* in the south (5:6). This seems to indicate that *D. acuta* has larger influence in the lower latitudes (below 52° north), while *D. acuminata* clearly dominates in the west, northwest, southeast and east above 52–52.5°. The data also showed an east to west axis, influenced by seasonal circulation patterns in the Celtic Sea, where the western Irish continental shelf follows a continuous anti-clockwise circulation pattern around the west of Ireland, which would support the idea that the *Dinophysis* species were moving from the south and towards the southwestern areas rather than towards the Southeast.

Figure 1 shows the overall patterns of the two main species in Ireland during the study period. *D. acuminata* appeared to be the dominant species most years, in 11 out of 13 years, but especially in 2005, 2012 and 2013, when the differences between the recorded species were quite significant. Interestingly, this pattern was reversed in 2014 and 2015, when the *D. acuta* records nearly double those of *D. acuminata*. This data agreed well with the toxin profile for the year, as DTX-2 was the principal toxin found in the shellfish. Moreover, although not shown in Figure 1, in 2014, *D. tripos* was also a dominant species, and it was recorded in 107 samples, which was significant taking into consideration that there have only been a total of 221 recorded samples for *D. tripos* since 2005. Thus, nearly 50% of all the records for *D. tripos* were observed in 2014, which coincides with the finding of quantifiable concentrations of pectenotoxins in shellfish in 2014 (data not shown here).

The seasonality trends of these three species are shown in Figure 2. *D. acuminata* appeared first in the water column and always peaked in the early summer months of June and July, while *D. acuta* generally appeared later in the summer and peaked a month later in July and August. *D. tripos* (data not shown in Figure 1) peaked in the late summer and early autumn months, in August and September, but the number of observations of this species was limited compared with the other two. Cells of these species were observed throughout the year, even during the winter months, but *D. acuminata*, in particular, can be observed in early spring, during March and April, in milder winters.

### 2.2. Shellfish Toxicity Data

DTX-2 was detected on an annual basis, and it was often observed above regulatory levels, particularly in mussels in the southwest. In 1994, prolonged closures of mussel harvesting areas were observed for the first time throughout the winter months into 1995. The persistence of a toxic event over the winter period and the next spring was not unusual, and it was determined by the timing and intensity of the toxic event. Figure 3 shows the concentrations of DTX-2 above the regulatory levels for the period from 2009 to 2017. Most years, DTX-2 concentrations were observed in late summer to early autumn and remained in shellfish tissues into the next spring, generally below the regulatory levels. The high DTX-2 amounts in 2014 and 2015 can be explained by the shift in dominance in the water column from *D. acuminata* to *D. acuta* (see Figure 1), when the trends were reversed from previous years. During years of low toxicity (for example, 2012, 2013 and 2017), DTX-2 was no longer present in shellfish tissues, and this data agreed with evidence of a decline of *D. acuta* observations in the water samples. The highest observed DTX-2 concentration was 7.63 µg∙g^−1^ in August 2010 in blue mussels (*Mytilus edulis*). DTX-2 above the regulatory levels has also been found in remainder tissues of king scallops (*Pecten maximus*) and Pacific oysters (*Crassostrea gigas*) (Table 3).

DTX-2 acyl derivatives were also found in the shellfish samples (Figure 4), and their concentration in the shellfish tissues was generally higher than that in the parent compound in the same sample, as comparisons between Figure 3 and Figure 4 would suggest. The highest ever observed hydrolysed DTX-2 concentrations were 7.84 µg∙g^−1^ in blue mussels and 7.1 µg∙g^−1^ in the remainder tissues of king scallops in October of 2014 (Table 3). Interestingly, DTX-2 esters have also been found to be above regulatory levels when converted back to their parent compounds through hydrolysis in samples of surf clams (*Spisula solida*) and cockles (*Cerastoderma edule*) (Table 3), even though the parent toxin has never been recorded for these species.

In summary, DTX-2 and its acyl derivatives were the predominant toxin compounds in Irish shellfish and were responsible for extended and prolonged closures during the winter months following the toxicity events when the highest concentrations (generally >1.5 µg∙g^−1^) occurred in September/October in extremely toxic years. When the highest concentrations (generally <1.5 µg∙g^−1^) occurred in August, this generally meant that there was no carry-over above the regulatory levels through to the following spring. Geographically, the southwest fared the worst with respect to harvesting closures, although this was not exclusive to this region, and other areas in the southern, western and northwestern coasts (Figure 5) were affected to a lesser extent. There is an inherent DSP toxicity bias towards the southwestern coast because of the predominance of longline rope mussel aquaculture production.

OA is found predominantly in blue mussels, where quantifiable concentrations were usually observed from May onwards and had generally increased to above the regulatory levels by June (Figure 6). These concentrations usually decreased below the regulatory levels by October–November, and the toxin was normally absent from the shellfish samples by the end of the year. The highest OA concentrations above the regulatory levels were observed in blue mussels (1.74 µg∙g^−1^ in June 2012) and in the remainder tissues of king scallops, 0.28 µg∙g^−1^ (Table 3). OA acyl derivatives were also present in shellfish (Figure 7) and followed the same concentration and seasonality patterns as the parent compound; however, acyl derivatives tended to stay longer in shellfish tissues during years of high toxicity. OA and its acyl derivative were predominant in the southwestern coast of Ireland (Figure 8).

In summary, OA and its esters resulted in concentrations above the regulatory levels on an annual basis and generally occurred earlier in the year than DTX-2 and its esters. The presence of OA and its accumulation in shellfish was normally associated with the presence of *Dinophysis acuminata* cells in the water column. 

Since 2012, the lipophilic method was modified to include the detection and quantification of PTX-1 and PTX-2, where the overall result is expressed as PTX equivalents µg∙g^−1^ (Figure 9). Since the monitoring began, quantifiable concentrations were rarely seen, and no PTX equivalent values have been observed above the regulatory levels. The highest value observed to date was 0.13 µg∙g^−1^ in the remainder tissues of king scallop. Interestingly, most of the quantified PTX equivalent values in the form of PTX-2 occurred in 2014 (from June–October). This coincided with high concentrations of DTX-2, DTX-2 esters and OA esters in shellfish, whilst PTX concentrations are recorded as being produced by *D. acuta*, *D. acuminata*, *D. fortii*, *D. caudata* and *D. norvegica*, and these phytoplankton species were occasionally observed in Irish waters. PTXs have more recently been thought to be produced by *D. tripos* [21,22], especially PTX-2. The 2014 data shows the largest number observations of *D. tripos* recorded since 2005 and the highest cell densities. The data strongly supports that *D. tripos* may have been responsible for PTXs being recorded in Irish shellfish samples for the first time since the monitoring of the PTX group began.

Maps of closures for harvesting per region during the period of 2009–2017 (Figure 10) suggest that the largest number of closures due to DSP incidents occurred in the southwest. In particular, the periods of 2009–2010 and 2014–2015 were exceptionally difficult, with protracted closures and high toxicity. These periods were related to the relative dominance of *D. acuta* in the water column, except for 2009, when more *D. acuminata* observations were made. The toxicity in these two periods showed toxic ‘overwintering’ (Table 4), especially in 2015 with 56 closures in January, 16 in February and 3 in March. During this time, there were no closures in the eastern and southeastern regions, and a small number of closures in the south, west and northwest. Figure 11 shows the total number of harvesting closures and the highest toxin OA equivalent recorded per year. As can be observed, 2014 and 2015 have been the worst two consecutive years for DSP toxins in Ireland since 2009 and 2010. 

## 3. Discussion

There is no doubt about the impact of the *Dinophysis* species and their associated toxins on the shellfish industry in Ireland over the years. The *Dinophysis* species, even at small cell concentrations in the water, can cause prolonged closures for the harvesting of bivalves in many areas of the country. The two main species associated with toxic events in Ireland are *Dinophysis acuminata* and *Dinophyis acuta*, although, as this review suggests, more diversity has been observed in the sampling areas, and in 2014, a toxic event in the south and southwest of the country also included another species, *Dinophysis tripos*, which was associated with the highest quantifiable concentration of pectenotoxins recorded. Pectenotoxins in Ireland have only been measured since 2012, and therefore, there is limited data for comparison, so this could be a case of increased monitoring rather than an increase in HAB events. In any case, future monitoring data will allow us to review this trend appropriately. This also highlights the importance of Phytoplankton monitoring, which through the proper identification of toxin-producing algae, can act as a valid warning system for the different toxin groups.

In 2014 and 2015, the biological data show a shift from *D. acuminata*-dominated samples to *D. acuta* domination, and this signalled a shift in the intensity and type of toxic compounds found in the shellfish for these years. DTX-2 became the dominant toxin in the shellfish, and this biological shift brought about the highest such toxin levels ever found in shellfish samples in Ireland. 

An interesting aspect of DSP episodes in Ireland is the ‘overwintering’ effect of toxins, especially in mussels, where there is a carry-over of toxins from one year into the next. Generally, these amounts are well below the regulatory level as winter approaches and tend to disappear completely during the spring bloom. This effect is more pronounced in the hydrolysed esters both for OA and DTX-2 than in the parent compound. DTX-1 is not a concern in Ireland, it has never been measured in any quantifiable amounts by LC-MS. Although we observe *Prorocentrum lima* in our samples, this benthic dinoflagellate has never been known to cause any toxic events in Irish waters.

DSP toxins are widely distributed in the country, as our harvesting closure plots suggest, but are particularly prevalent in the southwest. There are various reasons for this disparity between the regions; the prevalent oceanographic and weather conditions in the southwest and the movement of the Irish Coastal Current clockwise around the south and southwest of Ireland allow for dinoflagellates to aggregate inside the bays. Moreover, *D. acuta* is the dominant *Dinophysis* species in the south and southwest coasts compared to other areas, and it is more toxic than *D. acuminata.* Ultimately, longline rope mussels aquaculture are the most affected production areas in Ireland. 

Mussels seem to be a good biological toxin indicator in monitoring programmes, as they accumulate high levels of toxins, and according to our data, it appears to depurate slower than other bivalves. In other areas, for example the southeast (razor clams) and east (bottom mussels, razor clams, clams and cockles), there were no closures over this period. So, even after taking into consideration the differences in the number of sampling sites and shellfish species grown in the different regions, we can conclude with certainty that the southwest is the most affected region for DSP harvesting closures in the country. 

In conclusion, this review indicates that DSP is a prevalent toxin in shellfish that occurs on a regular basis in Ireland and causes huge economic loss to the industry; however, harvesting closures are required to protect human health. The data do not indicate a trend towards an increase or decrease of DSP events, but rather point more towards a cyclical trend of years with high toxicity interspersed with years with low toxicity, which can be clearly tracked through biological observations in terms of number of observations. The years with the highest toxicity are related to *D. acuta* dominance in the water column rather than *D. acuminata,* although *D. acuminata* is also involved in toxic episodes regularly. Sometimes, there is a combined effect of both species, especially in the southwest, where the ratio between these two species is similar. *D. acuta*, however, appears to be the more prominent of the species in the southwestern and southern region up to the 52–52.5° N in an east–west axis, with *D. acuminata* being dominant above this latitude in the east (Irish Sea) and west of the northwest region (Atlantic area), but *D. acuminata* is significantly stronger in the west of the northwest region. The aforementioned pectenotoxin event is probably an effect of increased monitoring and surveillance by the monitoring programme rather than an increase in HAB events. 

## 4. Materials and Methods 

### 4.1. Biotoxin National Monitoring Programme

#### 4.1.1. Sample Collection, Delivery and Lab Receipt

Samples of different species of marine bivalve molluscs are collected from aquaculture-classified production areas on a regular basis. The frequency is dependent on a number of parameters and can be increased or decreased due to the type of shellfish species, the time of year, the presence of known causative toxin producing phytoplanktonic species in the same or adjacent areas and the observation of quantifiable toxin concentrations observed in other shellfish species within the same production area. Generally, the frequency is weekly for mussels (*Mytilus edulis*), fortnightly for king scallops (*Pecten maximus*) and monthly for all other species (pacific oyster/*Crassostrea gigas*, flat native oyster/*Ostrea edulis*, surf clam/*Spisula solida*, razor clam/*Ensis siliqua* and cockles/*Cerastoderma edule*).

The majority of mussels produced in Ireland are done so through suspended longline rope mussel cultivation in the southwest and west. Bottom mussel cultivation also occurs to a lesser extent in a number of production areas in each of the different geographical regions. Oyster production is mainly via mesh bags on trestle tables and is mainly located along the southern, western and northwestern coasts. Razor clams are dredged mainly in the east and southeast. Dredged king scallops from classified production areas are mainly seasonal (October–March) in the southwest and harvested to a lesser extent in the west and northwest. There is a large, offshore scallop industry all around the coast of Ireland, as these scallops do not originate from classified production areas; the results from this dataset have been excluded from this review.

A minimum number of individuals of specific sizes for each shellfish species are collected, bagged, labelled and submitted to arrive at the Marine Institute laboratories the day after sample collection. Upon laboratory receipt, the sample details are logged into the HABs database, the shellfish are dissected to obtain the whole flesh (except scallop species) from each individual, which are pooled together and homogenised to obtain a whole flesh tissue homogenate weighing between 100–120 g. The sample homogenate is forwarded for toxin extraction and analysis, usually on the day of laboratory receipt. On occasions where analysis is not possible, the sample homogenate is either stored, refrigerated or frozen depending on when the analysis can be conducted. Scallop species originating from classified production areas are subjected to a different testing regime, in which the following tissues are dissected and pooled together from three different compartments: gonad (roe), posterior adductor muscle and the remainder tissues.

#### 4.1.2. LC-MS/MS Methodology

The method described here involves sample extraction with 100% methanol. A 2 g subsample of homogenate is twice extracted with 100% methanol (2 × 9 mL) and centrifuged. Free OA, free DTX-1, free DTX-2, PTX-1 and PTX-2 are determined by reverse phase liquid chromatography (LC) coupled with mass spectrometry (MS). A gradient method is applied to separate and elute the toxins in a single chromatographic run. A number of esters of the OA group are also analysed in this method, including 7-O-acyl esters and diol-esters. To determine these esters, an alkaline hydrolysis step is performed from 1 mL of the methanolic extract prior to analysis by LC-MS/MS. The step involves the addition of 125 µl NaOH, heating in a water bath to 76 °C for 20 min, and the addition of 125 µl HCl. This hydrolysis step converts any esters of the OA group toxins back to the original parent toxins OA, DTX-1 and DTX-2, which can then be quantitatively measured. Both pre and post hydrolysis extracts are analysed by LC-MS/MS. In recent years, the Marine Institute has validated the method to run on ultra-performance (UP) LC-MS/MS instruments. The UPLC tandem mass spectrometry method has been optimised and adapted from the methodology developed and validated by Dr. Arjen Gerssen [8]. The major changes to the published method are that the column used in this method is an Acquity BEH C18 2.1 × 100mm 1.75µm particle size, and the mobile phase B is not pH adjusted to 11. These changes were introduced to improve peak shape and repeatability. 

#### 4.1.3. Sample Results and Reporting

The lipophilic results for the DSP toxin group are usually available the day after sample receipt and are published on the HABs website (https://webapps.marine.ie/habs). The DSP results are reported as total OA equivalents in µg∙g^−1^ total tissue, which is the overall calculated hydrolysed result for OA and DTX-2 parent compounds. To calculate the results, a toxin equivalence factor of 0.6 is applied to DTX-2, a correction factor based on the % recovery of OA in certified reference material is applied to OA, and a dilution factor of 1.25 is applied to the calculation due to the hydrolysis step conducted on the sample.

For scallop samples, individual total OA equivalent results are reported for the individual dissected tissues: the gonad (roe), posterior adductor muscle and remainder (including the hepatopancreas) tissues. A calculated result for the whole tissue is also reported for scallop samples.

#### 4.1.4. Chemistry Data Review and Result Interpretation

For the purposes of this review, the data was extracted from the Marine Institute’s Harmful Algal Blooms (HABs) SQL Server database, which been in operation since 2001 and was used to record both bioassay (up to 2011) and chemical results for all the toxin groups for all the samples submitted for biotoxin analysis. The SQL queries were designed and run through a Microsoft Access front-end application developed in-house. For the purposes of this review, only the results from samples submitted from classified production areas have been reviewed and presented here. Sample results from offshore areas have been excluded from this review.

In total, for the years 2009–2017, 129,324 records were retrieved from HABs, where each record is an individual analysis for one of the compounds within the DSP group of toxins. This number of records equates to all the DSP analyses conducted on the 18,166 samples submitted in this time frame, which is just over 2000 samples per annum tested for DSP. The records from 2009–2017 were reviewed for OA, DTX-1 and DTX-2. Records for the esters of these parent compounds were available from May 2011–2017. For the PTX group, 33,699 records were retrieved, which equates to 11,079 samples analysed for PTX-1 and PTX-2, expressed as PTX equivalents. 

The records were reviewed to observe the seasonal occurrence and geographic distribution of the individual DSP isomers. All the toxin concentrations are expressed as µg∙g^−1^, and only quantifiable concentrations have been plotted in our figures. Any samples assigned values of <limit of detection or <limit of quantification have not been included in this review.

From an overall monitoring perspective and to assess the number of sites closed, data were also extracted from HABs to show the total number of samples that were above the regulatory levels, the site and species being assigned a ‘closed status’, and their geographic region during 2009–2017 to observe if the number of closures were increasing over this time period. These closure records only relate to the presence of DSP above the regulatory levels (>0.16 µg∙g^−1^ total OA equivalents) and do not include the closures in place during 2009–2017 attributed to the presence of other toxin groups Azaspiracid shellfish poisoning (AZP), Paralytic Shellfish poisoning (PSP) and Amnesic shellfish poisoning (ASP).

### 4.2. Phytoplankton National Monitoring Programme in Ireland

#### 4.2.1. Sample Collection, Transport and Delivery

Water samples are collected weekly for phytoplankton analysis in all shellfish production areas around the country. The Sea Fisheries Protection Agency (SFPA) is the governmental agency responsible for the coordination of sampling in these areas.

The samples are collected using a variety of methods, depending on the sampling site, tides and the type of shellfish grown in the area. Our preferred sampling method is the integrated sample using Lund tubes, but other techniques are also allowed, such as surface and discrete depth sampling using Niskin bottles. Generally, areas growing mussels on longline ropes use integrated sampling to sample the whole water column whereas in tidal sites with shallow depths and growing shellfish on the seabed, a surface or discrete depth sample is more pertinent at high tide. The Marine Institute furnishes samplers with all the required materials for collecting samples and training them on how best to take these and how to preserve, label and transport them safely. The Marine Institute in Ireland receives the samples generally one day after the sample collection.

#### 4.2.2. Sample Analysis

The Irish Phytoplankton programme uses 25 mL volume sedimentation chambers for water sample analysis, using the Utermöhl test method [14]. At least 12 h of settlement is necessary for this volume before the analysis commences. The Marine Institute uses inverted light microscopy with a range of objectives and optical properties to identify and enumerate the species found.

The full phytoplankton community analysis is carried out in over 40 sampling sites for a total of 85, covering all the bays around the country. The other 40–45 samples are analysed for toxic only species and the presence/absence of non-toxic ones. This means that all the toxic species, including all the *Dinophysis* genera, are identified in all the samples.

#### 4.2.3. Sample Results and Reporting

The samples are analysed within a day of sample settlement, and the results are reported regularly in our publicly available Harmful Algal Bloom (HAB) database. The results are available online within the same day of analysis. The cell densities are reported in cells per litre, and the genus *Dinophysis* is identified to the species level. 

#### 4.2.4. Harmful Algal Bloom (HAB) Database Phytoplankton Data Extraction

The HAB database was commissioned in November 2002, and for this review, the data from the period of 2005 to the end of 2017 was used. There is previous phytoplankton data available for Ireland in different formats before this date, going as far back as 1985 in previous databases.

The database was queried through Microsoft Access software and exported to Microsoft Excel spreadsheets. This review used data from 25,595 samples analysed and 346,186 species recorded during this period, 5980 of which were *Dinophysis* spp. 

In order to review all the data, the country was divided into different geographical areas: east, southeast, south, southwest, west and northwest in a clockwise direction instead of studying individual sites or bays. 

The eastern region extends to the north at Carlingford Lough (Cranfield House) in the frontier between the Republic and Northern Ireland in the Irish Sea, extends south to just above Wexford Harbour and includes the counties of Louth, Dublin and Meath. The southeast region starts at Cahore Point in Wexford bay and extend south towards the Celtic Sea to Wexford harbour, Rosslare, Waterford Harbour and Dungarvan to Helvick Head. The southern region extends from Helvick Head in Waterford to the west to Beacon Point to the Baltimore production area. This is mainly South County Cork including Youghal Bay, Cork Harbour, Oysterhaven, Kinsale and Roscarberry. The southwestern region starts in Baltimore and includes the southwest of Ireland, Counties Cork, Kerry and ends at Ballylongford. The western region starts at the Shannon Estuary in County Clare and extends northwards to Bellmullet Head at Dunanieran Point in Broadhaven North. The northwestern region commences here in North Mayo and extends northwards and eastwards to Lough Foyle in County Derry, on the border with Northern Ireland.

The biological data shown here has been normalised to account for the differences in the number of sites per geographical area, the sampling frequency and the number of years that samples have been collected. The best sites collect samples weekly all year-round and have been active since 2005, the starting point of our historical review. These conditions are not met by all the sites as these have a tendency to change overtime, where new sites start operating and others discontinue after many years. 

The east (8 sites), southeast (8 sites) and south (7 sites) have only a small number of production areas compared with the southwest (46 sites), west (60 sites) and northwest (38 sites). So, only the top sampling sites (5–10 sites) for each region, based on the conditions above, were chosen to compare the total number of observations and the ratios of *D. acuminata* and *D. acuta* for each area. 

To make this comparison, we first calculated the total number of samples per site by multiplying the average number of samples per year by the total number of years. The sum of all the samples collected over this period in the southern region sites was 1500, the lowest number in all the regions. The rule of three was used to calculate the total number of observations of *D. acuminata* and *D. acuta* in each region by multiplying 1500 by the total number of observations and dividing by the total number of samples for that region. Then, the ratio between the species was calculated using these values.

## Figures and Tables

**Figure 1 toxins-11-00061-f001:**
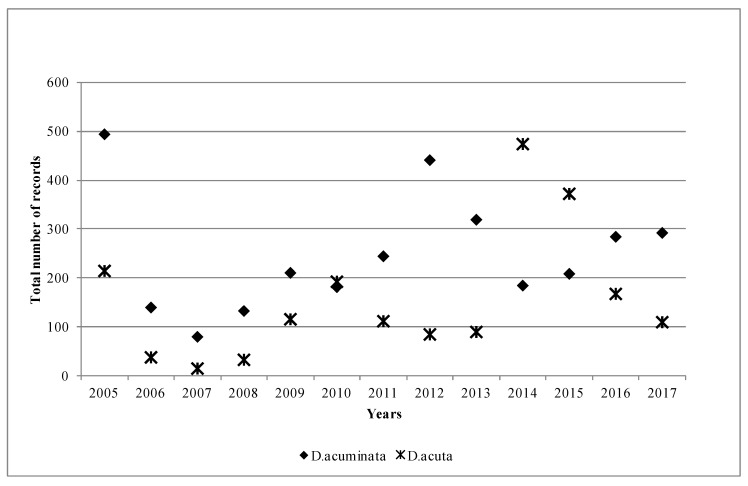
Number of observations of *Dinophysis acuminata* and *Dinophysis acuta* in Ireland by year between 2005 and 2017.

**Figure 2 toxins-11-00061-f002:**
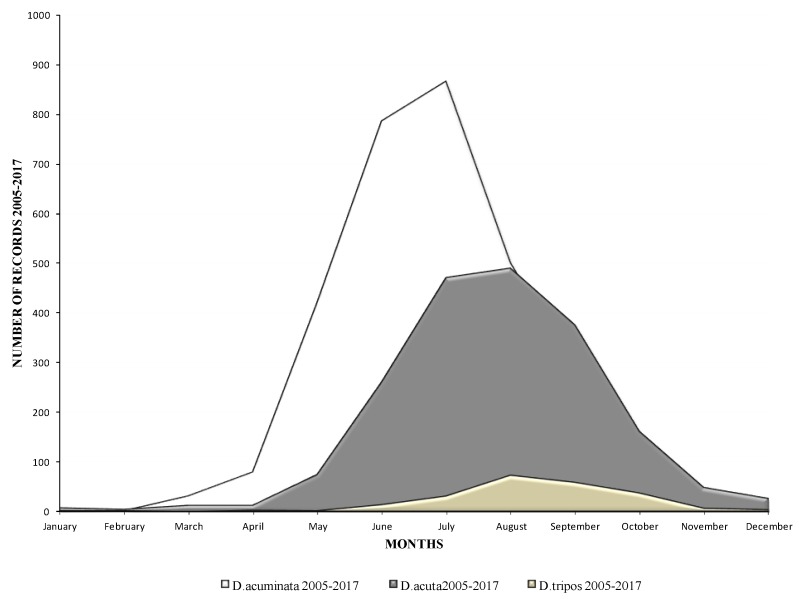
Seasonality of the species *D. acuta, D. acuminata* and *Dinophysis tripos* in Irish waters between 2005 and 2017.

**Figure 3 toxins-11-00061-f003:**
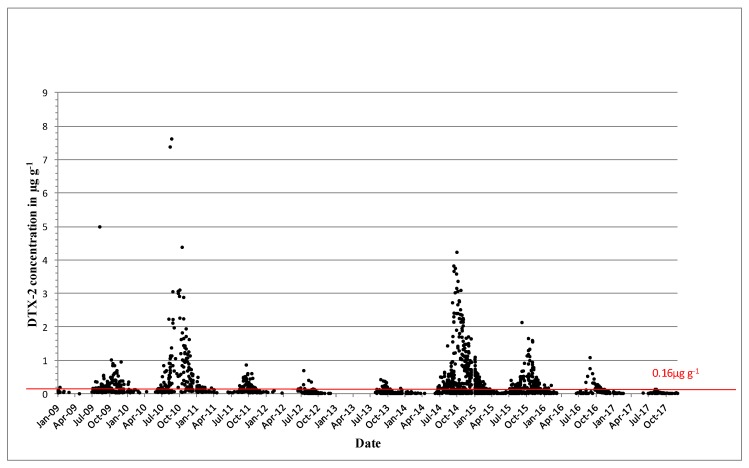
Dinophysistoxins-2 (DTX-2) quantifiable concentrations in µg∙g^−1^ found in shellfish between 2009 and 2017. The red line equals the closure level for diarrheic shellfish poisoning (DSP) toxin equivalents 0.16 µg∙g^−1^.

**Figure 4 toxins-11-00061-f004:**
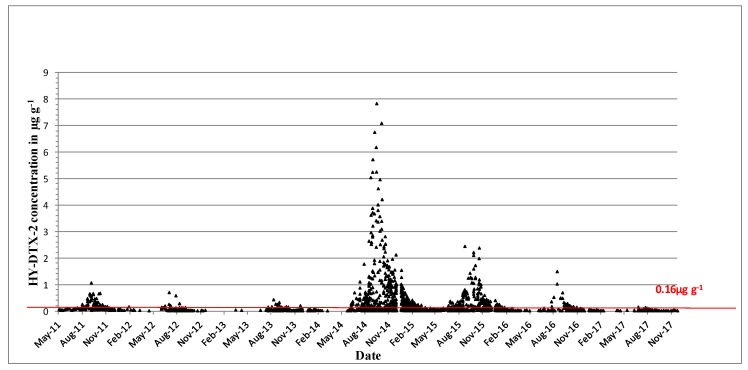
Hydrolysed DTX-2 quantifiable concentrations in µg∙g^−1^ found in shellfish in 2011–2017. The red line equals the closure level for DSP toxin equivalents 0.16 µg∙g^−1^.

**Figure 5 toxins-11-00061-f005:**
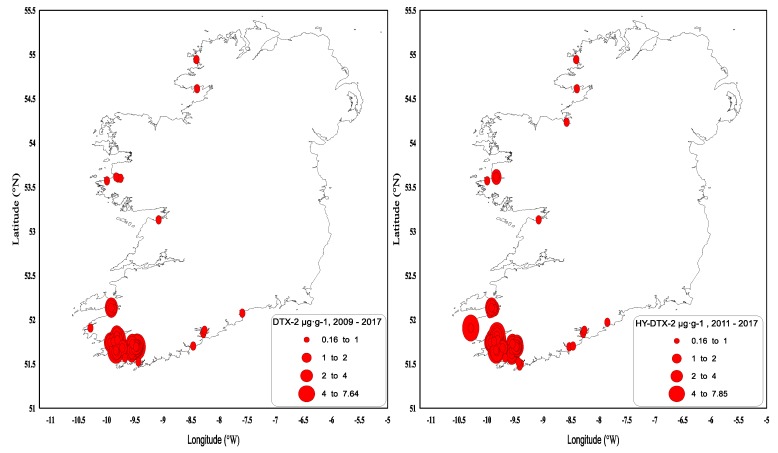
DTX-2 and hydrolysed DTX-2 values above the regulatory limit by geographical area in 2009–2017.

**Figure 6 toxins-11-00061-f006:**
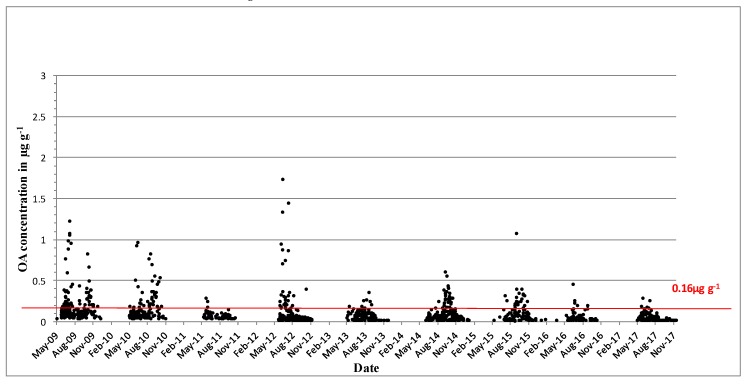
Quantifiable okadaic acid concentrations in µg∙g^−1^ found in shellfish in 2009–2017. The red line equals the closure level for DSP toxin equivalents 0.16 µg∙g^−1^.

**Figure 7 toxins-11-00061-f007:**
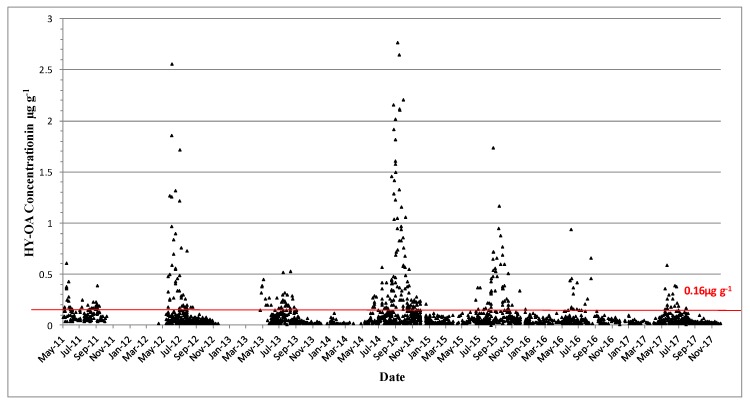
Hydrolysed okadaic acid quantifiable concentrations in µg∙g^−1^ found in shellfish in 2011–2017. The red line equals the closure level for DSP toxin equivalents 0.16 µg∙g^−1^.

**Figure 8 toxins-11-00061-f008:**
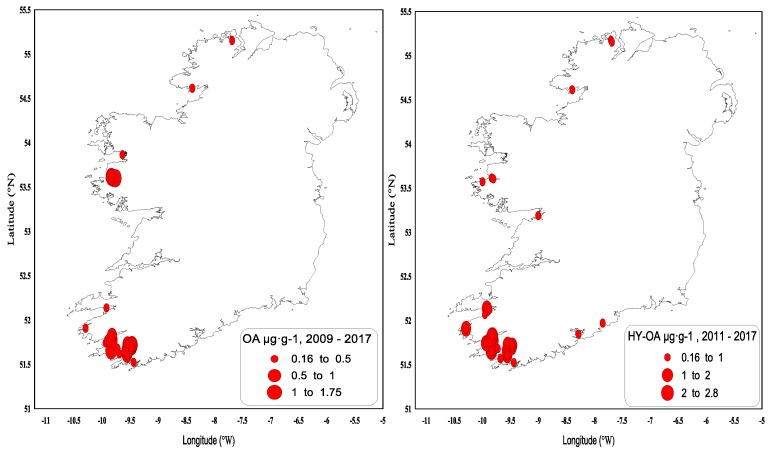
OA and hydrolysed OA values above the regulatory limit by geographical area in 2009–2017.

**Figure 9 toxins-11-00061-f009:**
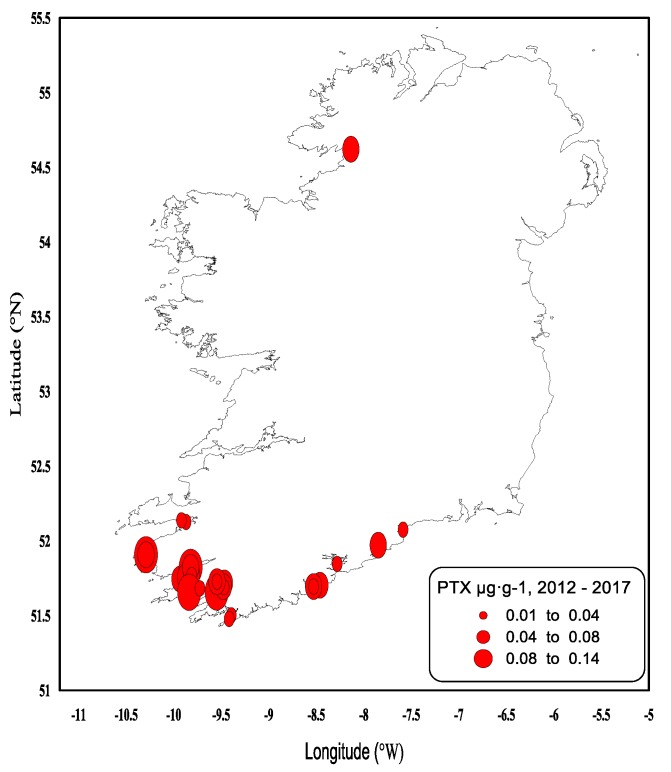
Quantifiable Pectenotoxin (PTX Equivalents) concentrations in shellfish in 2012–2017.

**Figure 10 toxins-11-00061-f010:**
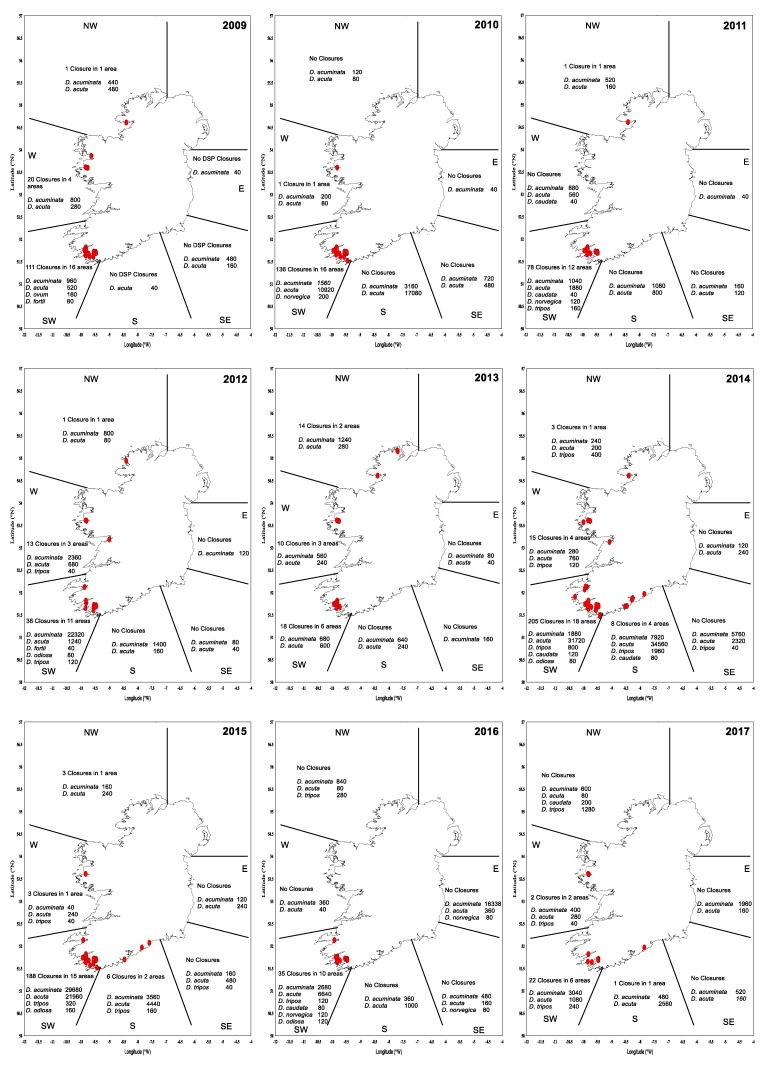
DSP Closure plots (2009–2017) by geographical area, *Dinophysis* diversity and maximum cell concentrations (cells∙L^−1^). Red dot (●) indicates a closure in a production area due to DSP detected above regulatory levels. NW: northwest; W: west; E: east; SW: southwest; S: south; and SE: southeast.

**Figure 11 toxins-11-00061-f011:**
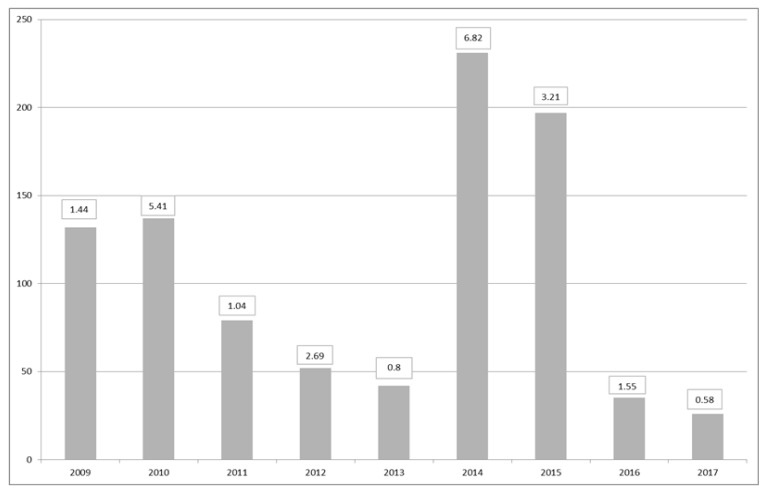
Number of harvesting closures due to DSP toxicity per year and maximum OA equivalents in each year in µg∙g^−1^ per year.

**Table 1 toxins-11-00061-t001:** Total number of observations of the *Dinophysis* species in Irish waters between 2005 and 2017 by geographical area (Note: *D. dens* is included here as synonym for *D. acuta*).

Geographical Areas							
Species Name	East	South East	South	South West	West	North West	Total Number of Observations
*Dinophysis acuminata*	160	103	126	1417	737	555	3098
*Dinophysis acuta*	109	51	141	1241	277	119	1938
*Dinophysis tripos*	0	2	9	151	13	46	221
*Dinophysis caudata*	0	0	1	12	1	3	17
*Dinophysis norvegica*	3	3	0	4	0	0	10
*Dinophysis odiosa*	0	0	0	10	0	0	10
*Dinophysis fortii*	0	0	1	7	1	0	9
*Dinophysis hastata*	0	0	0	0	6	1	7
*Dinophysis ovum*	0	0	0	3	1	0	4
*Dinophysis nasuta*	0	0	1	0	0	0	1

**Table 2 toxins-11-00061-t002:** Normalised data from 1500 samples per region.

Normalised, Estimated Number of Observations by Geographical Area			
Regions	*D. acuminata*	*D. acuta*	Ratio *D. acuminata/D. acuta*
East	134	91	0.68
Southeast	73	36	0.49
South	158	190	1.20
Southwest	345	288	0.83
West	196	73	0.37
Northwest	190	44	0.23

**Table 3 toxins-11-00061-t003:** Highest DSP toxin values observed in shellfish species in 2009–2017 expressed in µg∙g^−1^.

Species Name	Common Name	DTX-2 µg∙g^−1^	HY-DTX-2 µg∙g^−1^	OA µg∙g^−1^	HY-OA µg∙g^−1^
*Mytilus edulis*	Blue mussel	7.63	7.84	1.74	2.77
*Pecten maximus*	King scallop	0.27	7.1	0.28	1.92
*Crassostrea gigas*	Pacific oyster	0.36	0.29	<RL	<RL
*Cerastoderma edule*	Common cockle	<RL	0.75	<RL	0.34
*Spisula solida*	Surf clam	<RL	0.16	<RL	0.22

HY = Hydrolysed ester of the parent toxin; OA = okadaic acid; RL = Regulatory Level.

**Table 4 toxins-11-00061-t004:** Number of harvesting closures by region.

**Years**	**Southwest**											
	**Jan.**	**Feb.**	**Mar.**	**Apr.**	**May**	**Jun.**	**Jul.**	**Aug.**	**Sep.**	**Oct.**	**Nov.**	**Dec.**
2009	-	-	-	-	-	14	15	16	25	22	16	3
2010	2	-	-	-	4	13	16	28	15	20	28	10
2011	-	-	-	-	7	12	2	13	21	19	2	-
2012	-	-	-	-	-	12	15	9	2	-	-	-
2013	-	-	-	-	-	-	4	8	6	-	-	-
2104	-	-	-	-	-	10	12	23	49	34	41	36
2015	56	16	3	-	-	1	8	16	20	22	30	16
2016	4	-	-	-	-	8	4	5	6	8	-	-
2017	-	-	-	-	3	10	5	3	-	-	-	-
**Years**	**West**											
	**Jan.**	**Feb.**	**Mar.**	**Apr.**	**May**	**Jun.**	**Jul.**	**Aug.**	**Sep.**	**Oct.**	**Nov.**	**Dec.**
2009	-	-	-	-	-	5	13	2	-	-	-	-
2010	-	-	-	-	-	1	-	-	-	-	-	-
2011	-	-	-	-	-	-	-	-	-	-	-	-
2012	-	-	-	-	-	3	8	1	1	-	-	-
2013	-	-	-	-	-	-	1	6	3	-	-	-
2104	-	-	-	-	-	-	-	2	6	5	2	-
2015	-	-	-	-	-	-	-	3	-	-	-	-
2016	-	-	-	-	-	-	-	-	-	-	-	-
2017	-	-	-	-	-	-	-	2	-	-	-	-
**Years**	**Northwest**											
	**Jan.**	**Feb.**	**Mar.**	**Apr.**	**May**	**Jun.**	**Jul.**	**Aug.**	**Sep.**	**Oct.**	**Nov.**	**Dec.**
2009	-	-	-	-	-	-	-	-	1	-	-	-
2010	-	-	-	-	-	-	-	-	-	-	-	-
2011	-	-	-	-	-	-	1	-	-	-	-	-
2012	-	-	-	-	-	-	-	1	-	-	-	-
2013	-	-	-	-	-	7	3	4	-	-	-	-
2104	-	-	-	-	-	-	-	1	2	-	-	-
2015	-	-	-	-	-	-	-	-	-	-	-	-
2016	-	-	-	-	-	-	-	-	-	-	-	-
2017	-	-	-	-	-	-	-	-	-	-	-	-

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
