# Peer review of "Review of DSP Toxicity in Ireland: Long-Term Trend Impacts, Biodiversity and Toxin Profiles from a Monitoring Perspective"

_toxins, 2019, doi:10.3390/toxins11020061_

Round 1
Reviewer 1 Report
Manuscript ID: toxins-419396
Comments and Suggestions for Authors
The paper presents a review of the DSP toxicity data in Ireland from a monitoring perspective. The study examines historical data
(biological and chemical) of high value, searching for trends in this type of shellfish toxicity. The authors presented diversity of
phytoplankton species related with this toxicity and described toxin profiles. The results were reasonably presented and
conclusions are worth of publication. However, the manuscript must be improved before it can be recommended for publication.
Specific comments:
1. English should be thoroughly revised. Examples are the use of words that must be replaced as ”something” - line 38; “perhaps not surprisingly”
- line 43; “our waters” - line 112; “we don´t have as “- line 169; “our work”- line 283; “we should be” - line 287; “the pectenotoxin story” - line 324; “we are using” - line 395; “we have divided” - line 401; and word repetitions like in lines 68 and 186. Line 170, change “cells” to “Cells”
2. The title could be improved to be more attractive giving information on the results obtained. In my point of view results of oceanographic
conditions were not presented or discussed so a suggestion for the title is “ Review of DSP toxicity in Ireland: long term trend impacts, biodiversity
and toxin profiles from a monitoring perspective ”.
3. Abstract, Line 18. Please change the sentence order, the sentence “Also, the main toxic compounds ............in shellfish samples” could be
before the sentence “When D. acuta ......... prevalent toxin”
4. Key contribution. Please reformulate this section, the sentences are confuse and appearing with an arbitrary order, not coincident
with the importance in the study.
5. Introduction:
Line 35. Please take in consideration that in 2019 the study of marine biotoxins is not “recent”.
Line 51. please clarify the sentence about mouse bioassay or rat bioassay.
Line 88. Please change to another paragraph because the sentences refer no more to phytoplankton monitoring.
Line
92. Please add references to sustain the affirmation that “mussels are
not able to detoxify as fast as other bivalve molluscs”.
The same comment to line 306 (in discussion).
Line 120 to line 126. Here the authors summarizes some results, please change and replace by the objective of this review.
6. Results:
Line 136. Please clarify how was data normalisation obtained, how was the selection of the 1500 samples made? The same comment
for M and Methods (line 417), which were the conditions for the normalisation?
Line 164. Please add “.. . data not shown in figure 1”.
Line 173. This section should be “2.2. Shellfish toxicity data”
Line 174. Please change the sentence, it is too long.
Line
204. To better understand the differences between results of toxins in
long rope mussels and in other bivalve species, authors should
describe bivalve sampling in each geographical area (please add in Material and methods).
Lines 213 and 214 and lines 218-219- remove paragraphs
Line 225. “Its distribution generally follows that of Total DTX-2 equivalents…”- it is expressed in equivalents?
Line 232. Change sentence to “This coincided with high concentrations of DTX-2, DTX-2 esters and OA esters in shellfish, whilst PTX´s
are recorded as being produced by D. acuta, D. acuminata, D. fortii, D. caudata, D. norvegica and these phytoplankton species are occasionally
observed in Irish waters.”
Line 240. Please change the sentence to “Maps…………(figure 10) suggests that………”
7. Section 2.3. Figures and tables:
Table 1. Remove first line of the table, this information should be included in table caption.
Captions of Tables 1, 2 and 3. Captions don´t have enough information, examples: replace table 1 caption by “Number of records of
Dinophysis species in water samples collected between 2005-2017 by geographical area (note: D. dens is………of D. acuta”); add
information to caption of table 3 to explain differences between DTX2 and HY-DTX2.
Figure 1. Reduce the title of the figure; reduce the size of markers.
Figure 1 caption. Please add more information.
Figure 2. Please change the size of legends and colour of graph area to become similar to figure 1.
Figure 3. Caption is incomplete; change the scale of yy to better identify values near regulatory level; change µg/g to µg g-1.
Figure 9. In the figure legend is mentioned PTX µg/g, but in the caption is PTX1-2, please clarify.
Figure 10. Please add “….cell concentration (cells L-1)” to caption.
Figure 11. Change caption to “Number of harvesting closures due to DSP toxicity per year….”
8. Discussion
Line 279. Change the sentence to “There is no doubt about the impact of Dinophysis species and their…..”
Line 285. Please include references about phytoplankton species associated with PTX in other countries and compare with this study.
Line 306. Please add references to sustain the hypothesis of metabolization of lipophilic toxins.
9. Material and methods
The section “5.1. Bivalve sampling” is missing and should be included.
5.4. Please modify to “Water sample analysis” and add a reference to Utermohl method.
Author Response
Dear Reviewer,
Thanks for accepting to review our paper and for the comments and suggestions made. The recommendations were fair and comprehensive.
The text has now been fully revised and all the changes suggested have been made.
Regards

Reviewer 2 Report
The paper provides a useful long term overview of the phenomenon of DSP toxin contamination of shellfish in Ireland, it is closely aligned with
objectives of the special issue. The presentation and interpretation of the of the data is sound and it is worthy of publication. However, the paper
is spoiled because of the poor writing and the numerous errors of syntax and grammar in the text. It gives the impression that this is a hasty first
draft that has not had any proof reading. It is not suitable for publication in its present form as the text needs fairly extensive correction.
Examples include:
Line 13-14: “…suggest that we are not seen (seeing) an increase……”
Line 21: “ associated to (with) toxic events…”
Line 23: “..D. tripos on (in) that year.”
Line 24: The sentence “The harder hit areas by DSP outbreaks….” should read something like “ The areas of the country most affected by DSP
outbreaks are those practising long line mussel (Mytilus edulis) aquaculture”
The above is just in the abstract. The rest of the text is riddled with this sort of thing, too numerous to itemise individually here.
Overall the text is too wordy with numerous long tortured sentences. It could be made much clearer and more succinct.
For example, Lines 65-66: “The phytoplankton monitoring programme commenced a year after the rat bioassay in 1985 and its earlier version,
the main interest was on high biomass samples.” is a typical unclear sentence.
The text veers erratically between tenses in various places. Because it is reporting on historical data it should mainly be written in the past tense.
The manuscript needs to be corrected by an experienced editor.
Most of the figures need to have the borders removed. In Figs 5 & 9 the symbols have been squeezed so that they have become ovals unlike the
legend circles. These need correction.
Author Response
Dear Reviewer,
Thanks for accepting to review our paper and for the comments and suggestions made. I agree that the paper has been poorly and hastily written.
I hope that this new revision is a much improved version. The text has now been fully revised and all the changes suggested have been made. The
one comment we have not been able to comply with is the one about the symbols in figures 5 and 9, which look more oval than round. The maps
have been generated in Surfer and we found that in order to satisfy this demand, we would have to remove some of the data in these maps. There
are so many data points and the production areas in such a close proximity to each other than a lot of these points seem to be merging together.
I hope you take this into consideration.
Regards

Round 2
Reviewer 1 Report
The authors revised the manuscript appropriately according to the Reviewers comments. After careful consideration, I made a decision that the
manuscript is acceptable for publication in its present form.
Author Response
I would like to thank the reviewers for their work on reading, correcting and making useful comments on this article, which have helped us to improve
the paper substantially from the first draft.
Kind regards